# MicroRNAs in Plasma-Derived Extracellular Vesicles as Non-Invasive Biomarkers for Eosinophilic Esophagitis

**DOI:** 10.3390/ijms26020639

**Published:** 2025-01-14

**Authors:** Elena Grueso-Navarro, Leticia Rodríguez-Alcolado, Laura Arias-González, Ana M. Aransay, Juan-José Lozano, Julia Sidorova, Rocío Juárez-Tosina, Jesús González-Cervera, Alfredo J. Lucendo, Emilio J. Laserna-Mendieta

**Affiliations:** 1Department of Gastroenterology, Hospital General de Tomelloso, 13700 Tomelloso, Spainajlucendo@hotmail.com (A.J.L.); ejlaserna@sescam.jccm.es (E.J.L.-M.); 2Centro de Investigación Biomédica en Red de Enfermedades Hepáticas y Digestivas (CIBERehd), Instituto de Salud Carlos III, 28029 Madrid, Spain; 3Instituto de Investigación Sanitaria de Castilla-La Mancha (IDISCAM), 45071 Toledo, Spain; 4Department of Surgery, Medical and Social Sciences, Universidad de Alcalá, 28805 Alcalá de Henares, Spain; 5Instituto de Investigación Sanitaria Princesa, 28006 Madrid, Spain; 6Center for Cooperate Research in Biosciences (CIC bioGUNE), Basque Research and Technology Alliance (BRTA), 48160 Derio, Spain; 7Department of Pathology, Hospital General La Mancha Centro, 13600 Alcázar de San Juan, Spain; 8Department of Allergy, Hospital General de Tomelloso, 13700 Tomelloso, Spain

**Keywords:** eosinophilic esophagitis (EoE), extracellular vesicles (EVs), non-invasive biomarker, RNA sequencing, microRNAs (miRNAs), liquid biopsy

## Abstract

Eosinophilic esophagitis (EoE) is a chronic inflammatory esophageal disorder. The lack of non-invasive biomarkers currently results in dependency on endoscopy with biopsies for its diagnosis and monitoring. We aimed to identify potential non-invasive biomarkers using microRNAs (miRNAs) in plasma-derived extracellular vesicles (pEVs). This was a prospective single-center observational study of a discovery cohort of EoE patients (*n* = 26) with active disease (EoE.Basal) and after anti-inflammatory treatment (EoE.Post.tx) and control subjects (*n* = 16). Small-RNA-seq was performed to identify differentially regulated small RNAs (sRNAs). Candidate miRNAs were validated in an independent cohort (EoE patients, *n* = 33; controls, *n* = 14). The pEVs-sRNA cargo differed among conditions. Compared with controls, Ser_Comb_22, Leu_Comb_5, miR-10b-5p, and miR-125a-5p were upregulated in EoE.Basal, and miR-224-5p, miR-221-3p, let-7d-5p, and miR-191-5p were downregulated. The combination of miR-221-3p and miR-10b-5p showed the best diagnostic performance. Comparing paired EoE samples, miR-374a-5p and miR-30a-3p were upregulated in EoE.Basal, while miR-15a-5p and let-7d-5p were downregulated. Combined miR-30a-3p and miR-15a-5p showed the best AUC values, and miR-30a-3p alone was best as a monitoring biomarker (*p* = 0.001). In conclusion, pEVs-sRNA changed upon inflammation in EoE patients, and miR-30a-3p was proposed as a potential biomarker for monitoring the treatment. This study was the first to explore the use of pEVs as a non-invasive biomarker for EoE.

## 1. Introduction

Eosinophilic esophagitis (EoE) is a chronic immune-mediated disease, clinically manifested by symptoms of esophageal dysfunction and histologically characterized by an eosinophil-predominant inflammation limited to the esophagus [1]. An adaptive type 2 (T2) inflammatory response, mostly triggered upon exposure to specific food antigens, defines EoE as a particular form of food allergy that often co-occurs with other T2-related atopic diseases [2,3], as a late manifestation of the “atopic march” [4].

The incidence and prevalence of EoE continue to rise globally, with 6.3 new cases per 100,000 inhabitants annually and a prevalence of 60 cases per 100,000 inhabitants/year [5]. In fact, EoE has become the main cause of long-term dysphagia in children and young adults and ranks second for chronic esophagitis after gastroesophageal reflux disease (GERD) [6]. Thus, EoE is evolving into a significant healthcare challenge for the future, with associated health care costs matching those of inflammatory bowel disease [7].

Chronic or recurrent symptoms of EoE impair patient quality of life and that of their families [8]. They also produce elevated health care costs [9], largely due to the dependence on endoscopy with biopsies to establish the disease diagnosis (demonstrated by esophageal infiltration by at least 15 eosinophils per high power field (eos/hpf) [1]). Endoscopy with biopsies is also the only reliable method to monitor therapy effectiveness in patients with EoE [10]. The dependence on endoscopic procedures for the initial identification and management of cases hampers the early diagnosis, however. This delay is common and has been associated with the risk of progressive fibrous remodeling of the esophagus with stricture formation, thus aggravating the symptoms [11]. There is a need for reliable non-invasive biomarkers of disease diagnosis and activity in clinical practice as a surrogate for the peak eosinophil counts in esophageal biopsies [12]. Currently, biomarkers collected from peripheral blood, esophageal luminal secretions, and stool samples of EoE patients are too limited in their effectiveness to be incorporated into clinical practice, however [13].

Exploring the molecular cargo of extracellular vesicles (EVs) has emerged as a promising source of non-invasive biomarkers for inflammatory diseases [14]. The EVs, cell-derived nanometric lipid-bilayer vesicles, form a group of highly heterogenic entities that mediate juxtacrine, autocrine, paracrine, and endocrine cell–cell communication [15]. Consequently, EVs are abundant in several body fluids, including blood. Due to the molecular composition of EVs reflecting the status of the cell of origin [16], circulating EVs become peripheral messengers of a pathological process. The existing literature suggests that EVs might be a promising source for non-invasive biomarkers in EoE [17], but as yet, this has not been explored. Furthermore, the upregulation of vesicle-mediated transport in a proteomic characterization of EoE tissue has recently been reported [18].

Investigating EV-RNA is particularly compelling because RNAs encapsulated within EVs are shielded from enzymatic degradation in circulation. This protection enhances their detectability in liquid biopsies and promotes consistency and robustness across samples. In contrast, the presence of EV-RNAs alongside cell-free RNA and various cellular RNAs in whole blood samples introduces variable background noise, making it more challenging to identify molecular differences [19]. Small RNAs (sRNAs), and particularly microRNAs (miRNAs)—short (~20 nucleotides) non-coding sRNAs with a central role as posttranscriptional regulators of gene expression [20]—are involved in the regulation of allergic inflammation [21] and appear dysregulated in several esophageal diseases [22], therefore suggesting a contribution of miRNAs in esophageal disease progression. Indeed, several studies reported changes at the miRNA level in esophageal biopsies from EoE patients [23,24,25]. Previous attempts to find miRNAs as circulating biomarkers for EoE were inconclusive. Plasma levels of miR-146a, miR146b, and miR-223 were upregulated in EoE patients compared with controls [26], but no differences were found in miR-223 and other miRNAs (miR-203, miR-375, miR-142-3p, and miR-21) in serum samples from an EoE-pediatric cohort [25]. MiR-21 exhibited upregulation in biopsies and serum [26,27] and positively regulated eosinophil development [28]. However, it is often upregulated in allergic diseases, and no differences in miR-21 expression were reported in asthma and EoE plasma samples [27], diminishing its specificity for EoE. Others have failed to detect miRNAs differentially regulated in serum from EoE patients [23]. We hypothesized that the lack of reproducible findings may be attributed, in part, to inconsistent blood-processing methods [29] and enzymatic degradation affecting circulating RNAs, a challenge that could be surpassed by focusing on EV-encapsulated molecules [30].

Here, we explored the viability of using sRNAs from plasma-derived extracellular vesicles (pEVs) as non-invasive biomarkers for diagnosing EoE. Using a high-throughput method, we analyzed sRNAs in pEVs from EoE patients before and after anti-inflammatory treatment and compared them with control subjects. We then validated differentially regulated miRNAs using quantitative real-time PCR (RT-qPCR) in a new patient cohort and assessed the accuracy of various paired miRNA combinations in diagnosing the condition.

## 2. Results

### 2.1. Population of Study

As already described, the participants of this study were divided in two cohorts: a discovery cohort for the high-throughput transcriptomic analysis and a validation cohort to evaluate the relevant biomarkers (Table 1). The most common treatment among patients was swallowed topical corticosteroids (STCs), followed by treatment with proton-pump inhibitors (PPIs) and the four-food elimination diet (4-FED). Similar distribution of age, male percentage, time to post-treatment endoscopy and endoscopic features was observed in both cohorts. Gastroesophageal reflux disease (GERD) symptoms and dyspepsia were the most common reasons for the indication of endoscopy among control subjects.

### 2.2. Changes in RNA Cargo of pEVs Under Inflammatory Conditions

We first confirmed the enrichment of pEVs using size exclusion chromatography (SEC). The positive signal of CD9 and CD81 EV-associated tetraspanins (Figure 1A) confirmed the enrichment of pEVs in eluates (E)7 to E12, which was further confirmed when the signal faded after detergent lysis with Triton X-100 (Figure 1B). Eluates 7 to 12 concentrated the majority of pEVs with the less soluble protein content (Figure 1C). Finally, typical EV-like size distribution (Figure 1D) and shape entities appeared in pooled eluates (7 to 12) but were absent in negative controls (Figure 1E).

To explore whether transcriptomic signatures of pEVs could identify biomarkers in EoE, pEV-RNA was subjected to sRNA-seq and evaluated for quality control (QC) as a first step. By hierarchical clustering, using Pearson correlation as distance between samples and Principal component analysis (PCA), 11 samples were discarded as they were grouped together with three technical negative controls (Appendix A). Additionally, two more samples were discarded according to the results of the dendrogram obtained from unsupervised correlation analysis (Appendix A). Therefore, 47 samples were included in the bioinformatic analysis for the comparison of controls (*n* = 12) vs. EoE.Basal (*n* = 18) and the paired comparison of EoE.Basal vs. EoE.Post.tx (*n* = 17) (Appendix A). The reasoning for sample exclusion according to the QC analysis is described in Appendix A.

The transcriptome profile of pEVs from controls and EoE.Basal and EoE.Post.tx patients showed a heterogeneous pattern (Figure 2A) indicating few changes at the systemic level. Nonetheless, when analyzed by the PCA, the EoE.Basal group exhibited a relevant shift along the PC1 axis when compared with controls, which was partially reverted after treatment (EoE.Post.tx), thus indicating that the active inflammatory status characteristic of EoE.Basal has an impact on the sRNAs cargo of pEVs compared with the non-inflammatory status of controls and EoE.Post.tx samples (Figure 2B). Stronger differences were shown when controls were compared with EoE.Basal samples (Figure 2C) and when EoE.Basal was compared with EoE.Post.tx (Figure 2D).

Differential sRNA regulation between controls and EoE.Basal patients identified 757 sRNAs, of which 32 were significantly regulated (*p* < 0.05). The selection of predictive candidates (AUC > 0.75, deseq2 baseMean > 10, and *p* < 0.05) left eight sRNAs (six miRNAs and two tRNA-derived fragments (tRFs)). Of these, four sRNAs were upregulated in EoE.Basal pEVs (Ser_Comb_22, Leu_Comb_5, miR-10b-5p, and miR-125a-5p), while the other four were downregulated (miR-224-5p, miR-221-3p, let-7d-5p, and miR-191-5p) (Figure 3A,C, Table 2).

Paired comparison of EoE.Basal and EoE.Post.tx samples resulted in 797 differentially regulated sRNAs, of which 30 reached statistical significance (*p* < 0.05). Four miRNAs were identified as potential predictive candidates (Figure 3B,D, Table 3): two upregulated (miR-15a-5p and let-7d-5p) and two downregulated in EoE.Post.tx samples (miR-374a-5p and miR-30a-3p). It is worth noting that let-7d-5p was downregulated in EoE.Basal when compared with controls, while it was upregulated again after treatment (EoE.Post.tx), which suggested that let-7d-5p followed an expression pattern coincident with EoE activity, being downregulated upon inflammation.

### 2.3. Upregulated miRNAs in EoE Target Key Molecular Processes Related to EoE Pathophysiology

To decipher the biological properties and functions behind differentially regulated miRNAs, Gene ontology (GO) and Kyoto encyclopedia of Genes and Genomes (KEGG) analyses were carried out. As we intended to study specifically the upregulated functions in active EoE (EoE.Basal), the GO and KEEG analyses were restricted to the target genes of miRNAs upregulated in EoE.Basal compared with controls (miR-10b-5p and miR-125a-5p) and downregulated compared with EoE.Post.tx (miR-374a-5p and miR-30a-3p). The top 10 significantly enriched GO terms for biological process (BP), cellular component (CC), and molecular function (MF) are shown in Figure 4A. Regarding those upregulated in EoE.Basal when compared with control subjects, the target genes were found to mainly participate in cell proliferation and differentiation, miRNA transcription, and other transcription regulation categories. Interestingly, myeloid cell differentiation appeared as one of the most represented BP-GO terms, considering the pivotal role of eosinophils in EoE. Targets of miRNAs upregulated in EoE.Basal when compared with EoE.Post.tx patients agreed in four GO terms with the previous analysis: cytoplasmic stress granule, DNA-binding transcription factor binding, DNA-binding transcription activator activity/RNA polymerase II-specific, and ubiquitin-like protein ligase binding (Figure 4A). Both analyses showed similar overall functions, including cell proliferation/differentiation and gene expression, while adhesion-related GO terms were exclusively upregulated in EoE.Basal when compared with EoE.Post.tx.

KEGG analysis provided insights into the biochemical pathways upregulated in EoE.Basal pEVs. These seemed to be closely related to aspects of the pathophysiology of EoE, such as disruption of the epithelial barrier (HIF-1α, AMPK, and MAPK signaling pathways), inflammation (Wnt-signaling pathway), differentiation of immune cells (Th17 cell differentiation), and fibrosis (TGF-β signaling) (Figure 4B).

### 2.4. Combination of miRNAs Enhances the Diagnostic Performance as Biomarkers

We then studied whether the combination of selected miRNA candidates improved differentiation between conditions. The optimal diagnostic pair for EoE.Basal vs. controls was miR-221-3p and Leu_Comb_5 (AUC = 0.89), whereas the highest AUC (0.88) for a miRNA-only combination was observed with let-7d-5p and miR-221-3p (Figure 5A). For distinguishing EoE.Basal from EoE.Post.tx, the combination of miR-15a-5p and miR-30a-3p provided the highest AUC (0.90) (Figure 5B). Pairwise miRNA combinations revealed greater diagnostic performance than single pEV-miRNAs (Figure 5C).

### 2.5. Evaluation of Candidate miRNA Biomarkers in an Independent Validation Cohort

We applied RT-qPCR to confirm the differential expression of selected miRNAs in the validation cohort (EoE patients, *n* = 33; controls, *n* = 14). Based on the combinatory analysis, we tested let-7d-5p, miR-10b-5p, miR-221-3p, miR-15a-5p, and miR-30a-3p miRNAs. All miRNAs except for miR-221-3p had a similar expression trend to that in the sRNA-seq data (Appendix A), but only miR-30a-3p showed a significant dysregulation (*p* = 0.001) (Figure 6A).

The diagnostic power of suggested miRNA pairs in the validation cohort showed that miR-221-3p and miR-10b-5p in EoE.Basal vs. controls had the highest diagnostic accuracy (AUC = 0.70; 95% CI, 0.54–0.86). The combination of miR-15a-5p and miR-30a-3p was the best when comparing EoE.Basal vs. EoE.Post.tx (AUC = 0.71; 95% CI, 0.58–0.83) (Figure 6B). We also tested whether age could introduce any bias and restricted the analysis to adults (≥18 years; EoE patients, *n* = 24; controls, *n* = 11) (Appendix A), but no relevant changes were observed, except for the combination of miR-221-3p and miR-10b-5p that increased its diagnostic accuracy (AUC = 0.77; 95% CI, 0.59–0.95).

## 3. Discussion

The dependency on endoscopy with biopsies for the management of Eosinophilic esophagitis (EoE) has highlighted the need for non-invasive biomarkers in order to reduce the negative impact on patients’ quality of life and associated healthcare costs. Since previous attempts have not returned robust biomarkers, alternative sampling methods are needed to widen the search. To this end, we explored whether the plasma-derived Extracellular Vescicles (pEVs) might serve as biomarker carriers to indicate the inflammatory status in EoE patients. We observed differences in the small RNA (sRNA) cargo of pEVs, suggest different miRNA combinations with diagnostic potential, and postulate miR-30a-3p as a potential biomarker for monitoring the treatment of EoE. To the best of our knowledge, this constitutes the first-ever study to explore the role of EVs in EoE.

Analysis of the small-RNA landscape of pEV extracts showed a great heterogeneity among samples, due to its systemic origin, but enough to show global differences between inflammatory and non-inflammatory states. When plotted altogether, this suggested that pEV-sRNA was related to pathological dynamics since the cargo of EoE.Post.tx pEVs was more similar to that of the controls. While previous studies have documented transcriptomic alterations in biopsy specimens [26,31,32], the extent to which the RNA content of EVs mirrored that of tissue remained unexplored in this study. Nevertheless, the existing evidence indicates a concordance in miRNA expression across tissue, serum, and serum-derived EVs in hepatocellular carcinoma [33], with similar observations in chronic rhinosinusitis—a condition sharing histopathological features with EoE—where the transcriptomic analysis of mucus-derived EVs and matching biopsies showed overlapping differences [34]. A similar study could be performed in the context of EoE.

It should be noted that EVs were enriched from blood plasma to ensure optimal purification of vesicle entities but without further fractionation. Consequently, the resulting EV population comprised a heterogeneous mixture originating from various cellular sources, with platelet-derived EVs probably being the most abundant type, according to the literature [35]. The absence of further EV fractionation could have partly masked differences in the overall sRNA cargo between conditions. Here, the origin of EVs harboring biomarkers for EoE remains unknown. The contribution of tissue-specific EVs is unlikely, as they represent about 0.2% of total circulating EVs [36], but platelet-derived EVs themselves may reflect the inflammatory milieu as it was recently found that their lipidic and metabolic cargo correlated with the degree of inflammation in allergic patients [37]. Further studies will be needed to address this question.

Our study identified 32 sRNAs dysregulated in EoE compared with control subjects, along with 30 dysregulated sRNAs differentiating active EoE from treated EoE. We exclusively looked at the composition of miRNAs and tRNA derived fragments (tRFs) biotypes, both common components of EVs together with Y-RNAs and rRNAs [35] and involved in regulation of gene expression [38,39]. Although EV-tRFs might be regulated by T-cell activation [40], still little is known about their role in immune responses [41]. In contrast, the literature is extensive regarding the role of miRNAs, thus prompting our focus on miRNA candidates. In this study, we present a novel set of miRNA candidates from plasma EVs that diverge from previously proposed circulating miRNAs as biomarkers for EoE [26,27].

Moreover, it appears that identified miRNAs resemble the inflammatory activity in EoE as probing into the functional meaning of upregulated miRNAs in active EoE (miR-10b-5p, miR-125a-5p, miR-374-5p, and miR-30a-3p) revealed the regulation of genes involved in key molecular events for EoE. This included disruption of the epithelial barrier (AMPK [42], MAPK [43], and HIF-1α signaling [44]); inflammation (Wnt signaling [45]); fibrosis (TGF-β signaling [46]), which is particularly pronounced in untreated EoE patients [47]; and eosinophil degranulation (Th17 cell differentiation [48]). Of note, this functional overview was limited to the target genes of miRNAs overrepresented in EoE.Basal samples, so likely other important pathways involved in EoE pathophysiology (such as Th2 signaling) could be missed. Collectively, these findings suggest that pEVs could serve as informative mediators of some pathological processes underlying EoE.

Within the dysregulated EV-miRNAs, we selected those with greater AUC values for further validation. Although the RNAseq and RT-qPCR trends correlated, only miR-30a-3p showed significant upregulation in active EoE when evaluated by RT-qPCR. MiR-30a-3p is a member of the miR-30 family [49] and is involved in the regulation of tissue deve-lopment and the pathogenesis of various diseases. Its role has been studied in several inflammatory disorders [50,51,52], and it has been identified as a suppressor of eosinophilic inflammation in asthma [52]. While the role of miR-30a-3p in EoE remains unstudied, our literature review and miRNA target analysis indicated that miR-30a-3p targeted RUNX2 [52], CCR3 [53], and RTP4 (Appendix A)—three genes that are robustly upregulated in biopsies from patients with active EoE and serve as diagnostic markers [54,55]. Together, these findings suggest that miR-30a-3p may play a role in the pathology of EoE and warrants further investigation. Interestingly, another recent study found that miR-30a-5p is upregulated in the saliva of EoE patients [56].

The strengths of this study include the analysis of a cohort of patients representative of a standard EoE population, with newly diagnosed disease and inflammatory activity at baseline providing paired samples before and after effective anti-inflammatory treatment. Sample processing was carried out simultaneously to minimize technical variability, and validation of the results was carried out in independent EoE and control cohorts. We must also, however, recognize some limitations: the sample size could have been insufficient to detect some differences in miRNA expression as statistically significant; the EoE.Post.tx samples were considered homogeneous, and we did not consider the differential changes that could have been associated with certain particular therapies; and we have not performed a functional validation of the pathways identified in the computational prediction of targets of differentially expressed miRNAs. Finally, although the controls were matched for age and sex with EoE patients, we did not do so for concomitant atopies, and some observed differences could be a reflection of atopic inflammation in other organs beyond the esophagus, so we recommend including atopic controls in future studies addressing the search of biomarkers for EoE. Nonetheless, we reviewed the literature on similar studies conducted in common comorbid conditions associated with EoE, such as asthma [57,58], atopic dermatitis [59], and rhinitis [60,61]. Recognizing the limited number of methodologically comparable studies in these disorders, none of the differentially regulated miRNAs proposed in those studies were common to the potential biomarkers for EoE identified here. Only Li et al. [53] reported the downregulation of miR-30a-3p in peripheral blood from patients with asthma, which contrasted with our findings in pEVs.

## 4. Materials and Methods

### 4.1. Study Participants and Sample Processing

This was a prospective, single-center, observational study. Patients newly diagnosed with EoE were prospectively recruited between 2019 and 2022. The diagnosis of EoE was established following standard criteria [1], and plasma samples were obtained when active EoE (EoE.Basal) was confirmed by endoscopy and histology (≥15 eos/hpf). Patients were then treated according to clinical practice either by anti-inflammatory drugs, including proton-pump inhibitors (PPIs) and swallowed topical corticosteroids (STCs), or by food elimination diets (FEDs). Paired samples without inflammation (EoE.Post.tx) were obtained from patients who achieved histological remission of EoE (defined as <15 eos/hpf) after treatment. Individuals who underwent endoscopy during the same period, who showed no esophageal abnormalities and histologically normal esophageal biopsies, were recruited as control subjects in this study, selected to match patients with EoE in sex and age.

The study participants comprising 59 patients with EoE and 30 control subjects were randomly divided into two cohorts: the discovery cohort with 26 EoE patients and 16 controls and the validation cohort with 33 EoE patients and 14 controls (Table 1). The discovery cohort was processed for RNA-Seq to seek differential regulation of sRNAs between EoE.Basal vs. controls and EoE.Basal vs. EoE.Post.tx. The validation cohort served to evaluate differential regulation of identified candidates by RT-qPCR.

Blood samples were drawn from patients and control subjects during the endoscopic procedure. Blood was then centrifuged at 2000× *g* for 10 min to separate plasma, which was stored at −80 °C for further use. Endoscopic and histological evaluations were performed as previously described [62].

This study was approved by the Research Ethics Committee of Hospital La Mancha Centro in Alcázar de San Juan (study number: C-51, 30 November 2016). All patients and controls signed an informed consent form before sampling.

### 4.2. Purification and Characterization of EVs

Plasma EVs (pEVs) were isolated from 1 mL of plasma, thawed at 4 °C, and centrifuged at 5000× *g* for 30 min to remove protein aggregates and larger EVs. pEVs were purified using size exclusion chromatography (SEC) qEV columns according to the manufacturer’s protocol (Izon, Christchurch, New Zealand). The protein elution profile was measured directly on the void volume (3 mL) and the next 20 SEC eluates (0.5 mL each) using the Pierce BCA Protein Assay kit (Thermo Fisher Scientific, Waltham, MA, USA). Further characterization of EVs was performed using bead-based flow cytometry, electron microscopy, and nanoparticle tracking analysis (NTA). Bead-based flow cytometry was used on SEC eluates 1 to 15 to confirm the elution of EVs using the ExoStepTM Plasma/Serum kit (Immunostep, Salamanca, Spain). The void volume of SEC was used as the procedural control (PC). The aforementioned SEC eluates were concentrated up to 100 μL using a centrifugal filter Amicon ultra-0.5 10K (Merck Millipore, Burlington, MA, USA), of which equal amounts (50 μL/eluate) were mixed with CD9 capture beads and CD81-PE primary antibody, as indicated by the manufacturer. Isotype control capture beads were used as negative controls. After acquisition, SEC eluates were lysed with 0.1% Triton X-100 (TX-100) and acquired again. A loss of tetraspanin signal after TX-100 treatment indicated specific detection of EVs. EV-enriched eluates were pooled and concentrated using the centrifugal filter Amicon ultra-4 10K (Merck Millipore) for further RNA isolation. Qualitative characterization of EV size and shape was performed with electron microscopy and NTA. We have submitted all relevant data of our experiments to the EV-TRACK knowledgebase (EV-TRACK ID EV240152) [63].

### 4.3. RNA Isolation

Total RNA was isolated from EV samples using the Total RNA Purification Kit (Norgen Biotek, Thorold, ON, Canada) and treated with DNAse with RNAse-free DNAse I Kit (Norgen Biotek) to eliminate contaminant DNA, in accordance with the manufacturer’s protocol.

### 4.4. Transcriptomic Analysis by RNA-Seq

Seventy-one samples of total EV-RNA (26 EoE.Basal, 26 EoE.Post.tx, 16 control patients, and 3 technical negative controls) were used for small RNA-seq procedures at CIC bioGUNE (Derio, Vizcaya, Spain). DNase treatment was performed with a DNA-free™ Kit (Ambion, Foster City, CA, USA) in those samples where a peak of DNA was detected. Sequencing libraries were prepared following the manufacturer’s protocol for “NEXTflex™ Small RNA-Seq Kit v3” (Bioo Scientific Corporation, Austin, TX, USA). Libraries were then also quantified using a Qubit dsDNA HS DNA Kit (Thermo Fisher Scientific) and visualized on an Agilent 2100 Bioanalyzer using an Agilent High Sensitivity DNA kit (Agilent Technologies, Santa Clara, CA, USA) to assess their size distribution. The obtained libraries had the expected size (library average size: 200 bp), and their concentration was appropriate for sequencing in all samples except one from the EoE.Basal group. Therefore, 70 libraries were sequenced using a HiSeq2500 (Illumina, San Diego, CA, USA) with SBS (sequencing by synthesis) technology and single-end reads. Finally, sequencing data were converted into raw data (FASTQ files) for the analysis using Illumina’s package bcl2fastq.

Raw data files were deposited in NCBI’s Gene Expression Ommibus [64] and are accessible through GEO Series accession number GSE275710 (https://www.ncbi.nlm.nih.gov/geo/query/acc.cgi?acc=gse275710; accessed on 15 October 2024).

### 4.5. Bioinformatic Analysis

An initial approach of raw read processing included quality control analysis with the multiQC v1.6 tool, trimming for Illumina adapters, and alignment to the human genome using BWA (Burrow–Wheeler aligner) version 0.7.12 [65]. Gene quantification by means of HTseq software version 2.0.5 [66] was performed using gencode v.26 as an annotation database [67]. To determine the overall miRNA and transfer RNA-related fragment (tRF) composition for each sample, the miRge3.0 pipeline [68] from processed reads from no outliers against a set of sRNA annotation libraries was used to obtain their tabulated counts. Principal component analysis (PCA) was performed using princomp R function and plotted using ggbiplot R version 0.6.2.

Next, DESeq 2 [67] was employed to identify those small RNAs that were differentially regulated. In addition, the area under the ROC curve (AUC) was determined. For the selection of predictive sRNA as candidates in each comparison between groups, the following criteria were applied: AUC > 0.75, deseq2 baseMean > 10, and *p* < 0.05. The sensitivity, specificity, positive predictive value, and negative predictive value were also calculated for the selected candidates. Finally, SVM-RFE [69] was used to select the best combinations of two miRNAs to discriminate between the compared groups. To avoid overfitting, leave one out cross-validation (LOOCV) was applied. The AUC was calculated for each combination of candidates as an indicator of the diagnostic performance. All the analyses were performed using R statistical software (R-4.4.2). Functional analysis was performed by extracting the miRNA target genes from the Tarbase database [70]. The top 10 most significant pathways included in KEGG database from Enricher libraries [71] and the GO terms deleting redundant terms using the ClusterProfiler R Package version 3.10.1 [72] were then extracted.

Three technical negative controls were processed to detect any contaminants in the SEC columns and/or kits employed for pEV purifications and total RNA extraction. The first control sample was PBS buffer passed through a used SEC column. The second was PBS buffer passed through an unused SEC column. The third negative control was simply PBS buffer. All three negative controls underwent the same protocol for total RNA isolation.

### 4.6. cDNA Synthesis and RT-qPCR

A two-step RT-qPCR was performed for the validation of miRNA candidates. As a starting material, 2 µL of total RNA from pEVs was used for cDNA synthesis, using the TaqMan Advanced miRNA cDNA synthesis kit (Applied Biosystems, Foster City, CA, USA) according to the manufacturer’s protocol. A 1:10 cDNA dilution (5 µL) was employed in RT-qPCR and TaqMan Advanced miRNA assays (Applied Biosystems) for relative quantification: miR-10b-5p (ID#478494, Lot#P221212-008), miR-15a-5p (ID#477858, Lot#P230423-004), miR-30a-3p (ID#478273, Lot#P230208-008), miR-221-3p (ID#477981, Lot#P230606-006), let-7d-5p (ID#478439, Lot#P230726-006), let-7a-5p (ID#478575, Lot#P221122-008), and TaqMan Fast Advanced Master Mix II (Applied Biosystems) as described in the manufacturer’s protocol. qPCR was run on a QuantStudio™ 3 (Applied Biosystems) in triplicates at 95 °C for 20 s, followed by 40 cycles at 95 °C for 1 s and 60 °C for 20 s. Let-7a-5p was used for the normalization of miRNAs in EV samples based on sRNA-seq data (let-7a-5p ranked best according to highest mean expression levels, lowest standard deviation, and coefficient of variation in a set of sRNAs with fold-change values between −1.2 and 1.2).

### 4.7. Statistical Analysis

The demographic and clinical data were expressed as a median with an interquartile rank (IQR). For comparisons between groups for the RT-qPCR results, we first tested the normal distribution of samples with the Saphiro–Wilk test. As none of the groups passed the normality test, non-parametric statistical tests were used. The Mann–Whitney U test was used for the comparison of the control vs. EoE.Basal, and the Wilcoxon signed-ranked test was used for paired samples (EoE.Basal vs. EoE.Post.tx). Relative expression data were displayed as the mean ± the standard error of the mean (SEM). The area under the ROC curve (AUC) was used as the indicator of diagnostic performance. Statistical analysis was performed using GraphPad Prism software version 9 (GraphPad Software, San Diego, CA, USA).

## 5. Conclusions

In summary, we explored the transcriptomic landscape of plasma EVs in EoE. We provided the first evidence that circulating pEV cargo varies according to the inflammatory activity in these patients and that RNA cargo associates with key molecular mechanisms involved in the pathophysiology of EoE. Furthermore, we suggested several pairwise combinations of candidate miRNAs that could enhance their diagnostic performance and postulated miR-30a-3p as a potential monitoring biomarker for the treatment of EoE. Validation of these findings will require future investigations involving larger cohorts, while also considering the influence of concomitant atopic conditions. In conclusion, this study establishes a foundation for further research exploring the role of EVs in EoE.

## Figures and Tables

**Figure 1 ijms-26-00639-f001:**
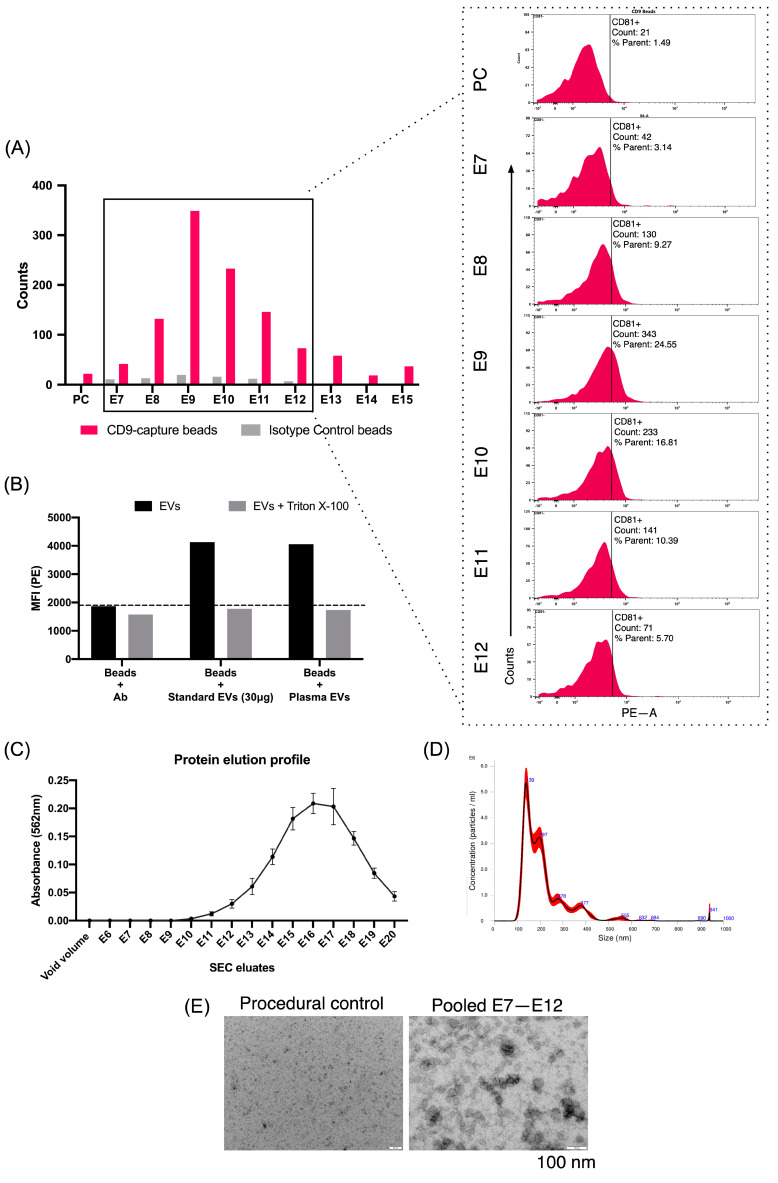
Characterization of plasma-derived extracellular vesicles (pEVs). (**A**) Evaluation of the size exclusion chromatography (SEC) elution profile by bead-assisted flow cytometry. Pink bar graphs show the number of CD81^+^ events along the eluates of SEC column. Isotype control coated-beads are shown as gray bars. The phycoerythrin (PE) signal corresponding to CD81^+^/CD9^+^ events is displayed in the histograms (**A**, right panel) of eluates 7–12 and the procedural control. The percentage of PE^+^ (CD81^+^) events is indicated within the histogram plots. PC, procedural control. (**B**) Bar graph showing the PE median fluorescence intensity (MFI) values before (black) and after (gray) treatment with 0.1% Triton X-100. (**C**) Protein elution profile of SEC eluates by the bicinchoninic acid assay (BCA) method. (**D**) Representative image of pEVs size distribution analyzed by nanoparticle tracking analysis (NTA). (**E**) Transmission electron microscopy (TEM) images of negative control and pool of eluates E7–E12 showing the presence of EV-like particles.

**Figure 2 ijms-26-00639-f002:**
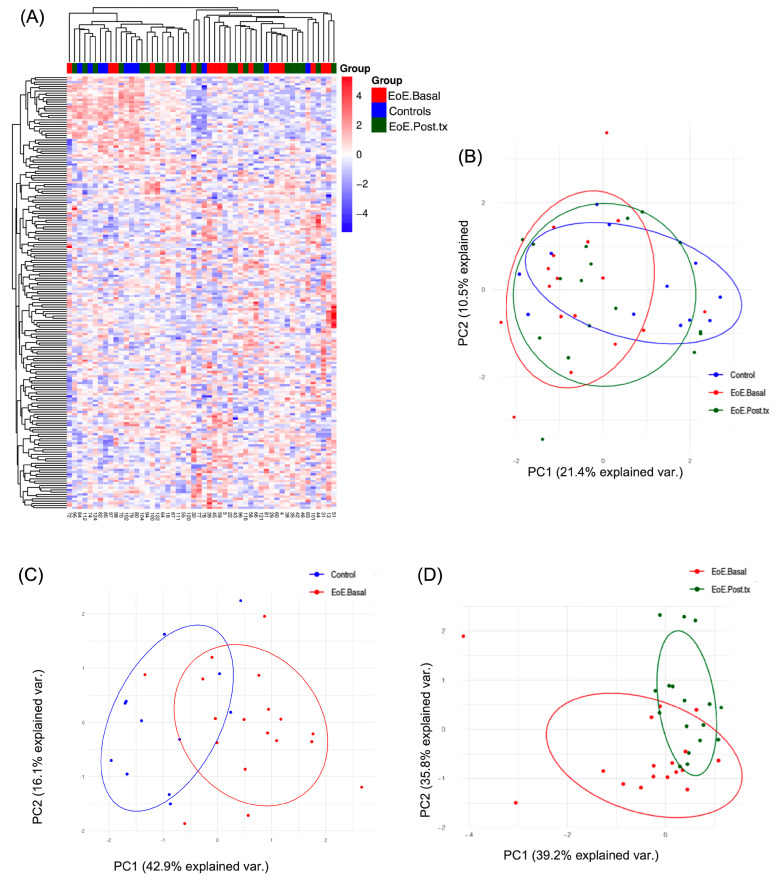
Landscape of the sRNA profile in pEVs of controls and EoE groups in the discovery cohort. (**A**) Heat map showing the sRNA expression profile of controls, EoE.Basal, and EoE.Post.tx. (**B**) PCA comparison of controls and EoE.Basal and EoE.Post.tx patients. (**C**) PCA of controls vs. EoE.Basal and (**D**) PCA of paired comparisons EoE.Basal vs. EoE.Post.tx. Controls, *n* = 12; EoE.Basal, *n* = 18; and EoE.Post.tx *n* = 17.

**Figure 3 ijms-26-00639-f003:**
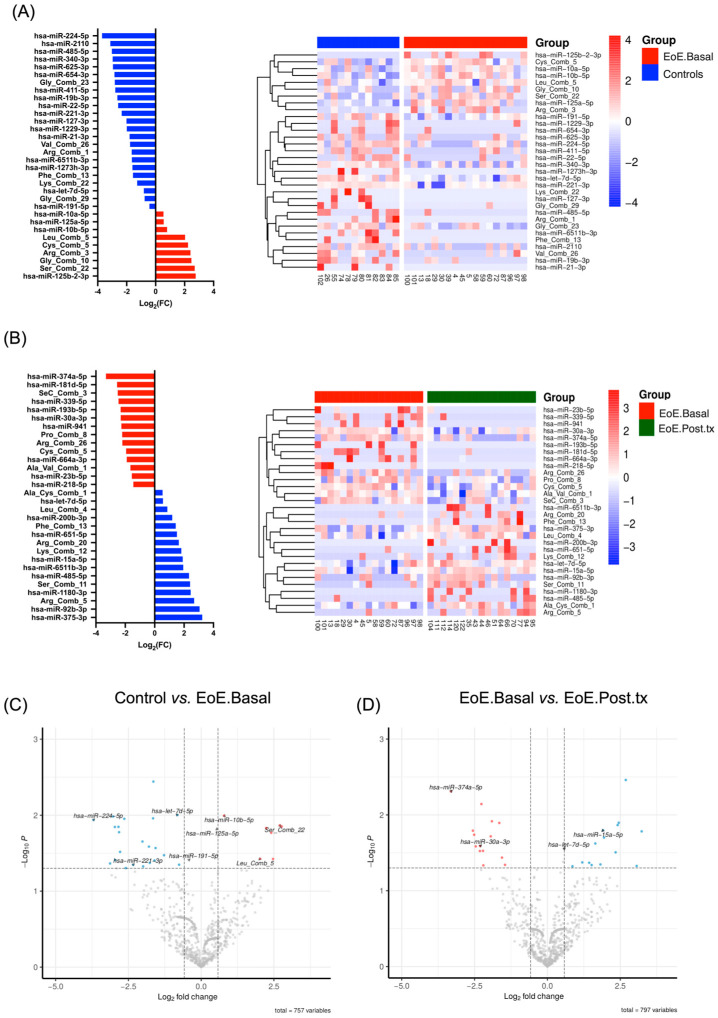
Identification of differentially regulated small RNAs. (**A**) Diverging bar charts (left) and heat map (right) showing the sRNA signature according to their fold-change value (*p* < 0.05) co- rresponding to the comparisons controls vs. EoE.Basal and (**B**) EoE.Basal vs. EoE.Post.tx. (**C**) Volcano plots depicting significantly altered sRNAs between controls vs. EoE.Basal and (**D**) EoE.Basal vs. EoE.Post-tx. Colored dots represent sRNAs with fold-change values ≤ −1.5 or ≥1.5. Highlighted sRNAs exhibit AUC > 0.75. Upregulated sRNAs in EoE.Basal are shown in red, while downregulated sRNAs in EoE.Basal are shown in blue. Controls, *n* = 12; EoE.Basal, *n* = 18; and EoE.Post.tx, *n* = 17.

**Figure 4 ijms-26-00639-f004:**
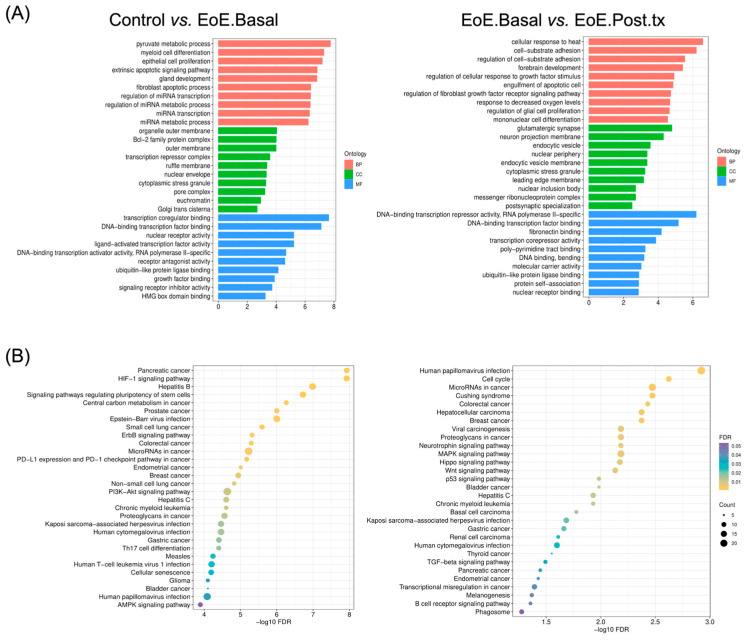
Functional analysis of differentially expressed miRNAs upregulated in EoE.Basal. (**A**) Gene Ontology (GO) analysis for biological process (BP) in red, cellular component (CC) in green, and molecular function (MF) in blue. The horizontal axis represents the number of genes that contribute to a particular item. (**B**) Bubble plots showing the enriched KEGG pathways of the miRNA target genes upregulated in EoE.Basal when compared with controls or EoE.Post.tx.

**Figure 5 ijms-26-00639-f005:**
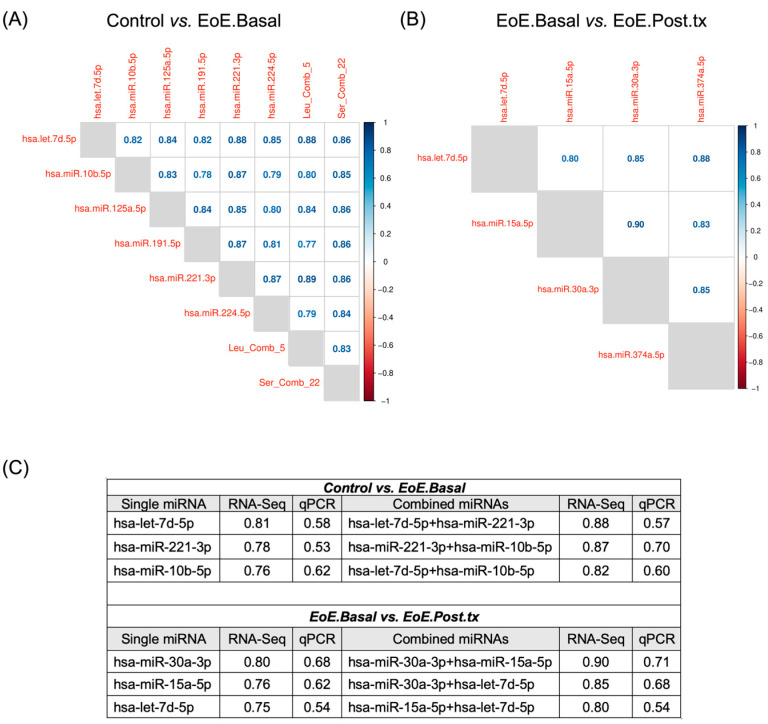
Diagnostic performance of the combination of miRNA pairs. Correlation matrices displaying AUC values of evaluated miRNA combinations for comparison of (**A**) controls vs. EoE Basal and (**B**) EoE.Basal vs. EoE.Post.tx. (**C**) Comparison of AUC values for single miRNAs and for paired miRNA combinations in both discovery and validation cohorts.

**Figure 6 ijms-26-00639-f006:**
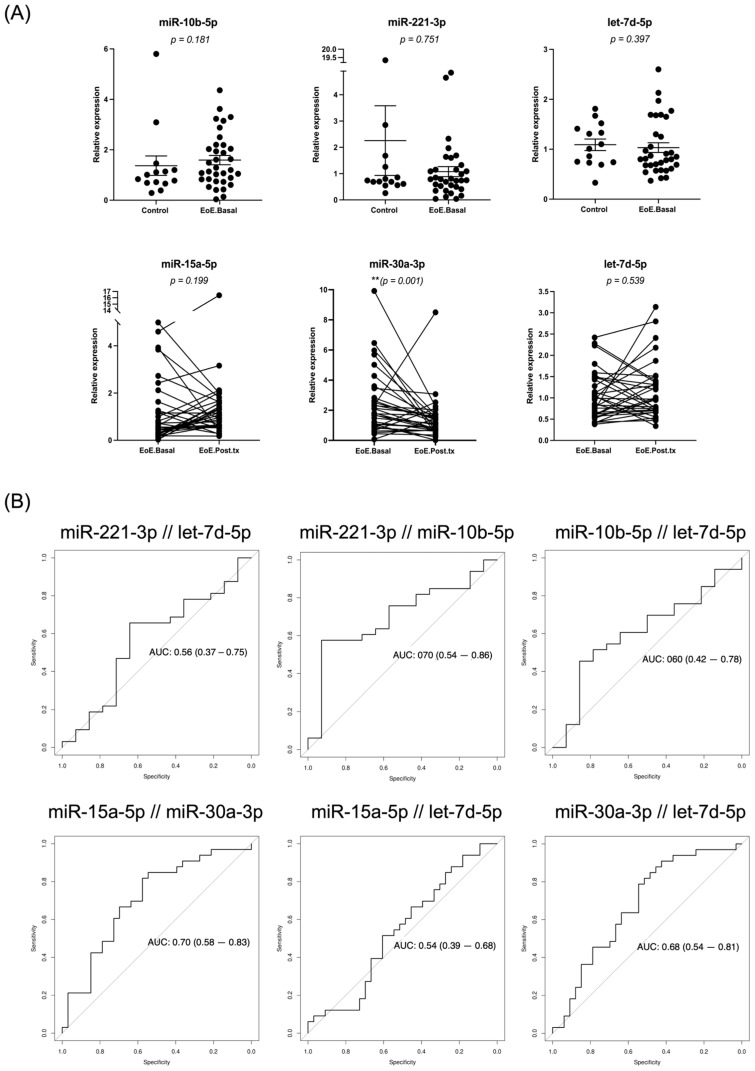
Validation of miRNA candidates by RT-qPCR. (**A**) The expression of five miRNAs from RNAseq analysis, miR-10b-5p, miR-221-3p, let-7d-5p, miR-15a-5p, and miR-30a-3p, was performed on the validation cohort (controls, *n* = 14; EoE.Basal, *n* = 34; EoE pairs (EoE.Basal and EoE.Post.tx), *n* = 33). (**B**) ROC curves of the selected miRNA combinations among those with the best AUC values for control vs. EoE.Basal and EoE. Basal vs. EoE.Post.tx comparisons.

**Table 1 ijms-26-00639-t001:** Main demographic and clinical characteristics of the patients with EoE and control subjects included in the discovery and validation cohorts.

	Discovery Cohort		Validation Cohort	
	EoE Patients	Control Subjects	EoE Patients	Control Subjects
N	26	16	33	14
Age, median (IQR)	23.5 (16.0–37.5)	32.5 (20.0–40.8)	30.0 (17.0–44.5)	39.5 (21.5–44.3)
Male, *n* (%)	21 (81)	12 (75)	21 (64)	9 (64)
Treatment, *n* (%)	3 (12) 4-FED 5 (19) PPI 18 (69) STC	*	3 (9) 4-FED 4 (12) PPI 26 (79) STC	*
Days between basal and post-tx endoscopies, median (IQR)	94 (84–179)	*	89 (52–182)	*
Peak of eosinophils/hpf (basal), median (IQR)	54 (34–90)	0 (0–0)	54 (35–85)	0 (0–0)
Peak of eosinophils/hpf (post-tx), median (IQR)	0 (0–4)	*	0 (0–0)	*
EREFS (basal), median (IQR)	4 (3–5)	*	4 (3–5)	*
EREFS (post-tx), median (IQR)	0 (0–0)	*	0 (0–0)	*
Main reason for endoscopy, *n* (%)	*	2 (12) epigastric pain 3 (19) dysphagia 4 (25) dyspepsia 7 (44) GERD	*	1 (7) epigastric pain 1 (7) other 3 (21) dysphagia 4 (29) GERD 5 (36) dyspepsia

4-FED, four-food elimination diet; EREFS, endoscopic score system measuring edema, rings, exudates, furrows, and strictures; GERD, gastroesophageal reflux disease; hpf, high-power field; IQR, interquartile range; PPI, proton pump inhibitors; STC, swallowed topical corticosteroids; tx, treatment; (*), not-applicable.

**Table 2 ijms-26-00639-t002:** Differentially regulated miRNAs and tRFs that met the prefixed criteria for candidate selection in the comparison between control subjects and EoE.Basal patients.

	FC	*p*	AUC	SN	SP	PPV	NPV
hsa-let-7d-5p	−1.76	0.01	0.81	0.72	0.83	0.87	0.67
hsa-miR-224-5p	−13.00	0.01	0.78	0.94	0.58	0.77	0.88
hsa-miR-221-3p	−5.06	0.04	0.78	0.78	0.67	0.78	0.67
Leu_Comb_5	4.08	0.04	0.78	0.83	0.75	0.83	0.75
Ser_Comb_22	6.50	0.01	0.76	0.89	0.58	0.76	0.78
hsa-miR-10b-5p	1.73	0.01	0.76	0.89	0.58	0.76	0.78
hsa-miR-125a-5p	1.47	0.02	0.76	0.67	0.92	0.92	0.65
hsa-miR-191-5p	−1.35	0.04	0.75	1.00	0.50	0.75	1.00

A positive fold change (FC) corresponds to upregulation in EoE.Basal patients, ordered from higher to lower AUC values. *p*, *p*-value; AUC, area under the ROC curve; SN, sensitivity; SP, specificity; PPV, positive predictive value; NPV, negative predictive value.

**Table 3 ijms-26-00639-t003:** Differentially regulated miRNAs that met the prefixed criteria for candidate selection in the comparison between EoE.Basal and EoE.Post.tx.

	FC	*p*	AUC	SN	SP	PPV	NPV
hsa-miR-30a-3p	−4.96	0.03	0.80	0.94	0.59	0.70	0.91
hsa-miR-374a-5p	−9.93	0.00	0.78	1.00	0.47	0.65	1.00
hsa-miR-15a-5p	3.76	0.02	0.76	0.71	0.71	0.71	0.71
hsa-let-7d-5p	1.48	0.03	0.75	0.82	0.65	0.70	0.79

A positive fold change (FC) corresponds to upregulation in EoE.Post.tx samples, ordered from higher to lower AUC values. *p*, *p*-value; AUC, area under the ROC curve; SN, sensitivity; SP, specificity; PPV, positive predictive value; NPV, negative predictive value.

## Data Availability

Raw data files of RNA sequencing were deposited in NCBI’s Gene Expression Ommibus and are accessible through GEO Series accession number GSE275710 (https://www.ncbi.nlm.nih.gov/geo/query/acc.cgi?acc=gse275710; accessed on 15 October 2024). In addition, relevant data of the methodology employed for extracellular vesicle isolation from blood plasma were submitted to the EV-TRACK knowledgebase (EV-TRACK ID EV240152).

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
