# Peer review of "MicroRNAs in Plasma-Derived Extracellular Vesicles as Non-Invasive Biomarkers for Eosinophilic Esophagitis"

_ijms, 2025, doi:10.3390/ijms26020639_

Round 1

Reviewer 1 Report

Comments and Suggestions for Authors

I appreciate the opportunity to review the manuscript for publication in MDPI IJMS. The manuscript provides valuable insights into the potential of microRNAs (miRNAs) within plasma-derived extracellular vesicles (pEVs) as non-invasive biomarkers for Eosinophilic Esophagitis (EoE). The manuscript is well written and organized. The study uses a combination of RNA sequencing and RT-qPCR validation, which strengthens the reliability of the results. Identifying miR-30a-3p as a potential monitoring biomarker might contribute to EoE management. I have a few comments as follows.

Heterogeneity in EV Sources: The lack of fractionation of EV populations may introduce variability due to the diverse cellular origins of EVs. This might mask disease-specific signals. Further characterization of EV subtypes is of great value.

Control Group Limitations: The control group is not matched for concomitant atopic conditions, which could confound the findings. The authors should discuss the possibility to include atopic controls to isolate EoE-specific changes.

Discussion in Functional Insights: While miRNAs were identified, limited functional experiments were conducted to confirm their roles in EoE pathophysiology. Discussion in functional validation is necessary to corroborate the computational predictions. Further, only miR-30a-3p showed significant dysregulation in the validation cohort. This raises concerns about the reproducibility of other miRNA candidates.

Author Response

We are grateful for the thoughtful feedback on our manuscript. We appreciate that your

comments addressed three key aspects of improvement and discussion on our

manuscript. We hope we satisfied your concerns with a more detailed explanation of

the mentioned aspects. Below, you will find our detailed responses along with the

corresponding revisions implemented in the updated manuscript.

Reviewer 1:

I appreciate the opportunity to review the manuscript for publication in MDPI IJMS.

The manuscript provides valuable insights into the potential of microRNAs (miRNAs)

within plasma-derived extracellular vesicles (pEVs) as non-invasive biomarkers for

Eosinophilic Esophagitis (EoE). The manuscript is well written and organized. The

study uses a combination of RNA sequencing and RT-qPCR validation, which

strengthens the reliability of the results. Identifying miR-30a-3p as a potential

monitoring biomarker might contribute to EoE management. I have a few comments

as follows.

Heterogeneity in EV Sources: The lack of fractionation of EV populations may

introduce variability due to the diverse cellular origins of EVs. This might mask

disease-specific signals. Further characterization of EV subtypes is of great value.

Thank you for this comment, which addresses a really important issue. Indeed,

the great heterogeneity of EVs could be important to enhance the signal from a

candidate biomarker. However, to tackle EV heterogeneity from a biomarker discovery

phase is tricky. For example, we do not know where our potential candidate biomarkers

will be, and fractionation of the different EV-types (including variable sizes, densities,

or even surface markers) would require the use of multiple isolation methods such as

density gradient isolation by ultracentrifugation, microfluidics, affinity-based capture,

etc. Considering that the isolation methods of choice introduce variability in the

detection of the EV cargo, it is key to consider a scalable isolation method for

biomarker discovery from the beginning that could be transferred into the clinical

practice in further steps. The following references are examples of papers in which

circulating biomarkers were found without fractionation of EVs: PMID: 19011622,

31413567, 32627209.

Control Group Limitations: The control group is not matched for concomitant atopic

conditions, which could confound the findings. The authors should discuss the

possibility to include atopic controls to isolate EoE-specific changes.

This is also an important point. Thank you for pointing it out. Certainly this

represents a limitation of our research which was mentioned in lines 341-343 of the

manuscript. The inclusion of atopic patients among the control group would have

reinforced the reliability of our results, but we find it difficult to include these atopic

controls as they are not normally submitted to endoscopy in absence of esophagealrelated

symptoms. Recruiting atopic patients without esophageal symptoms to form

part of a control group would have been very complicated in a proof-of-concept study,

and we would have also had difficulties explaining it to an ethics committee. In addition,

EoE is the main cause of esophageal symptoms in young atopic subjects, with which

the subjects would become part of the experimental group. Therefore, we could only

consider control subjects as those with absence of eosinophils in the esophagus or

any other esophageal abnormality which could be checked in endoscopy performed

due to non-esophageal symptom. After our results, we are now identifying new atopic

patients submitted to endoscopy, to confirm our results in a future study. We agree

with the reviewer that this is an issue to be taken into account for future studies looking

for EoE-specific biomarkers. We have added a sentence to make clear this

recommendation in the discussion section (lines 344-345).

Additionally, we have evaluated that this issue could have a minor impact in our result

by checking the following:

• We could not identify similarities in differentially regulated miRNAs in studies

methodologically comparable to ours including patients with atopies often

concomitant to EoE. This was mentioned in the discussion section in lines 345-

350.

• We performed a correlation clustering map including presence of atopies

(among other characteristics) and miRNA expression (figure below). Samples

from non-atopic patients did not cluster together, indicating negligible effect of

atopies in the miRNA-EV cargo.

• We confirmed that the collection of samples did not specifically coincide with

peaks of atopic burden, as most concomitant atopies in EoE have a seasonal

nature. Distribution shown in the histogram plot below.

Discussion in Functional Insights: While miRNAs were identified, limited functional

experiments were conducted to confirm their roles in EoE pathophysiology. Discussion

in functional validation is necessary to corroborate the computational predictions.

Further, only miR-30a-3p showed significant dysregulation in the validation cohort.

This raises concerns about the reproducibility of other miRNA candidates.

Thanks again for this comment. The main objective of this work was to explore

the potential utility of EVs in blood as a biomarker of EoE activity. Here, the functional

analysis was merely thought to investigate the connection between EV cargo and EoE

activity, and give us a hint about their link. Whether the overrepresented functional

pathways were directly linked to EoE pathophysiology could not be concluded from

this work because it was out of our scope to demonstrate a functional role of the

candidate biomarkers, but rather identify and evaluate their robustness as diagnostic

and monitoring options. In the new version of the manuscript, we have included this

lack of functional validation as a limitation of our study (lines 339-341).

Reviewer 2 Report

Comments and Suggestions for Authors

In this manuscript, the author identified non-invasive biomarkers-miRNAs from plasma-derived extracellular vesicles in EOE patients and explored their therapeutic use in diagnosing EOE. They reported that pEVs-sRNA changes upon inflammation in EoE patients, and miR-30a-3p proposed as a potential biomarker for monitoring the treatment. Have a few comments/questions/suggestions:

Line 115. Please provide the full term of SEC for the first time use.

Figure 1. Please include a control histogram of CD81-PE in the right panel of figure 1A to show how the positive gating was set.

Figure S1. The author performed PCA analysis for all samples, identified 11 outliers and excluded them for the following analysis. However, PCA analysis could be easily distorted by outliers when outliers were not removed. To exclude that many samples, please use other outliers detection methods to verify that those samples are true outliers.

Figure 2B. Please include the description of the groups by colors in the legend. Same as the legend shown in figure S1B.

Figure 2B. Line 150-152. EOE basal showed shift from PC1 towards to PC2 compared to control. And treatment reverted that trend. Please explain how the interpretation of PC1 and 2 related to inflammatory status were concluded.

Line 201. And downregulated compared to EOE.POST.tx.

Figure 4 and Line 218. KEGG analysis showed th17 differentiation was involved. How about th1/th2, tregs? Would expect to see th1/th2, tregs differentiation in a "type 2" immune response. If not, can you provide insights in the discussion?

Line 348-349. In this study, plasma samples were obtained when active EoE was confirmed by endoscopy and histology. The dependence on endoscopic procedures delays the early diagnosis of EOE as mentioned in the introduction. Has author tested the mi-rna at early time point before the endoscopy? If so, how early would you expect to see the difference compared to healthy or control?

Line 365. Please use g force instead of rpm.

Author Response

We appreciate the constructive feedback provided on our manuscript. Your comments

have contributed to improving the quality and depth of our work. Please, find below

the detailed responses and corresponding corrections in the revised version of the

manuscript.

Reviewer 2:

In this manuscript, the author identified non-invasive biomarkers-miRNAs from

plasma-derived extracellular vesicles in EOE patients and explored their therapeutic

use in diagnosing EOE. They reported that pEVs-sRNA changes upon inflammation

in EoE patients, and miR-30a-3p proposed as a potential biomarker for monitoring the

treatment. Have a few comments/questions/suggestions:

Line 115. Please provide the full term of SEC for the first time use.

Thank you very much for your careful review of our manuscript. The full

meaning of SEC has been added in the new version of the manuscript in lines 116-

117.

Figure 1. Please include a control histogram of CD81-PE in the right panel of figure

1A to show how the positive gating was set.

We appreciate this comment. Changes added in “Figure 1A_edited”.

Figure S1. The author performed PCA analysis for all samples, identified 11 outliers

and excluded them for the following analysis. However, PCA analysis could be easily

distorted by outliers when outliers were not removed. To exclude that many samples,

please use other outliers’ detection methods to verify that those samples are true

outliers.

Thanks again for your comment. Here we employed different visualization

methods for detection of potential outliers including the three analyses plotted in

Supplementary Figure 1 (heatmap, PCA and clustering). In all of them, the outliers

correlate well with the negative controls and we applied a very strict criteria to remove

any possible outlier sample to avoid their influence in further analyses. Attending to

the reviewer´s request we have evaluated the outliers using the MDQC method (PMID:

17933854), which is based in Mahalanobis distance (MD), and we have obtained 11

outliers, the 3 negative controls and 8 samples that were coincident with previous

analyses.

Figure 2B. Please include the description of the groups by colors in the legend. Same

as the legend shown in figure S1B.

Changes added in “Figure 2B_edited”.

Figure 2B. Line 150-152. EOE basal showed shift from PC1 towards to PC2 compared

to control. And treatment reverted that trend. Please explain how the interpretation of

PC1 and 2 related to inflammatory status were concluded.

The interpretation is based in the inclusion criteria for patients within the

EoE.Basal group, which was the presence of inflammation described as infiltration of

more than 15 eosinophils per high power field in the esophagus biopsies, while for

controls the biopsies had to be negative for eosinophilic infiltration. Considering these

inclusion criteria, we interpreted the differences between groups in Figure 2B

according to their inflammatory status. This has been clarified in lines 153-154 in the

new version of the manuscript.

Line 201. And downregulated compared to EOE.POST.tx.

It has been corrected in the new version of the manuscript in line 205.

Figure 4 and Line 218. KEGG analysis showed th17 differentiation was involved. How

about th1/th2, tregs? Would expect to see th1/th2, tregs differentiation in a "type 2"

immune response. If not, can you provide insights in the discussion?

We appreciate this important comment. Considering the well stablished Th2

nature of EoE, it is difficult to speculate an explanation for the absence of Th1/Th2 or

Tregs pathways in our KEGG analysis, but it seems unlikely that Th1/Th2 or Tregs are

not involved. Likely, as the functional analysis was limited to the miRNAs

overrepresented in EoE.Basal samples, we could think that the target genes of these

miRNAs are not preferentially involved in these pathways. In any case, the functional

analysis here did not intend to demonstrate the pathophysiology of EoE but to link, at

least partially, the sRNA cargo of circulating EVs with the disease. This limitation has

been recognized in the new version of the manuscript in lines 311-314.

Line 348-349. In this study, plasma samples were obtained when active EoE was

confirmed by endoscopy and histology. The dependence on endoscopic procedures

delays the early diagnosis of EOE as mentioned in the introduction. Has author tested

the mi-rna at early time point before the endoscopy? If so, how early would you expect

to see the difference compared to healthy or control?

In the frame of this project, we collected the plasma samples strictly at the time

of the endoscopic procedure, therefore we could not evaluate changes in potential

candidates at early time points. We may speculate that as EoE patients derived to

Gastroenterology outpatient clinic had already symptoms suggestive of EoE when

endoscopy was requested (and normally performed two or three months after in our

center), probably the differences between them and healthy controls could be detected

when esophageal symptoms became evident.

Line 365. Please use g force instead of rpm.

It has been corrected in the new version of the manuscript in line 373.
